# The Analysis of the Mycobiota in Plastic Polluted Soil Reveals a Reduction in Metabolic Ability

**DOI:** 10.3390/jof8121247

**Published:** 2022-11-25

**Authors:** Marta Elisabetta Eleonora Temporiti, Lidia Nicola, Carolina Elena Girometta, Anna Roversi, Chiara Daccò, Solveig Tosi

**Affiliations:** Laboratory of Mycology, Department of Earth and Environmental Sciences, University of Pavia, 27100 Pavia, Italy

**Keywords:** soil, fungi, biodiversity, metabarcoding, plastic, pollution, metabolic profile

## Abstract

Plastic pollution is a growing environmental issue that results in its accumulation and persistence in soil for many decades, with possible effects on soil quality and ecosystem services. Microorganisms, and especially fungi, are a keystone of soil biodiversity and soil metabolic capacity. The aim of this research was to study soil fungal biodiversity and soil microbial metabolic profiles in three different sites in northern Italy, where macro- and microplastic concentration in soil was measured. The metabolic analyses of soil microorganisms were performed by Biolog EcoPlates, while the ITS1 fragment of the 18S ribosomal cDNA was used as a target for the metabarcoding of fungal communities. The results showed an intense and significant decrease in soil microbial metabolic ability in the site with the highest concentration of microplastics. Moreover, the soil fungal community composition was significantly different in the most pristine site when compared with the other two sites. The metabarcoding of soil samples revealed a general dominance of *Mortierellomycota* followed by *Ascomycota* in all sampled soils. Moreover, a dominance of fungi involved in the degradation of plant residues was observed in all three sites. In conclusion, this study lays the foundation for further research into the effect of plastics on soil microbial communities and their activities.

## 1. Introduction

Plastic pollution is a peculiar problem of the modern world. Only in 2020, 367 million tons of plastic was globally produced, and it was estimated that the cumulative global production has reached 10 billion tons since 1950 [1,2]. Moreover, it is reported that 80% of plastic waste ends up directly or indirectly in natural environment, where they can be fragmented into smaller pieces as a result of the action of natural agents [3,4]. Plastic particles < 5 mm are defined microplastics [5], which are reported to range from 62.5 to 40,800 per kg of dry soil [6,7]. This concentration is four to 23 times higher than the concentration in the ocean [8,9]. The presence of plastics and microplastics can affect soil quality by altering both chemical and physical properties, such as pH, water-holding capacity, electrical conductivity, porosity, nutrient content [3,10], and microbial communities [3,11].

Soil microbial communities, their compositions, abundances, and activities are essential for soil quality [12,13]. In particular, fungi enable the circulation of organic and inorganic compounds, producing enzymes involved in decomposition processes that enable biomass recycling [14,15,16]. Moreover, they play a key role in plant growth promotion, enhancing soil fertility and releasing secondary metabolites and bioactive compounds [17]. Changes in the soil environment can highly affect its microbial components, which respond with different metabolic intensities and activities [3,18,19]. Indeed, they vary in response to external disturbing factors, including plastics, resulting in the primary biological indicators of soil modifications [19,20,21,22]. These alteration in fungal communities, metabolism and enzymes activities related to plastic introduction in soil were investigated mainly with in vitro studies. The research carried out by Fan and colleagues [3] showed alterations at the fungal community level, with an increase in Ascomycota and reduction in Simpson and Shannon indexes. Another study reports an increment of *Mortierella* with the increasing of microplastic addition but without going so far as to influence soil fungal diversity (Shannon index, and Chao1 index) [23]. In addition, the effect on the microbial (fungal and bacterial) activity is controversial. Indeed, in some cases, a stimulation of enzyme activities is reported with a positive correlation with the increase in plastic pollution [3,20,21,24]. In other studies, an opposite effect was observed, showing the inhibition of extracellular enzyme activities [19,22,25,26].

The soil fungal communities in plastic-polluted environments have not been often studied in depth, using next-generation sequencing. However, several studies were performed on communities’ modification related to other pollutants. The presence of contaminants in soil ecosystems reduces microbial diversity, richness, and population size as well as microbial activities [27,28]. For example, in vivo studies reported an increase in Ascomycetes in soils contaminated by hydrocarbons or heavy metals [27,29,30].

The aim of the present research was to analyse soil fungal communities and the metabolic profile of three different sites, where macro- and microplastic concentration in soil was measured. The perspective was to assess how the impact of emerging pollutants such as plastics can alter fungal communities and their activities, affecting soil quality.

## 2. Materials and Methods

### 2.1. Study Site and Sample Collection

The study sites were in the province of Pavia, in the Po Valley, Northern Italy. The first site (45°32′00.62″ N; 9°21′81.79″ E; 90 m.a.s.l.) is an agroecosystem near an anthropized area where a great quantity of plastic waste was found just below the soil surface (here called the Polluted AgroEcosystem—PAE; Appendix A). This site is characterised by wild grass and some shrubs (mainly *Robinia pseudoacacia*).

The second site (45°31′67.43″ N; 9°22′11.2″ E; 90 m.a.s.l.), located near the first one, is also an agroecosystem, but no plastic waste was visually detected during sampling (here called AgroEcosystem—AE; Appendix A). The vegetation is dominated by *R. pseudoacacia* shrubs.

The third site is inside the “Bosco Siro Negri”, which is an Integral Nature Reserve (here called INR; Appendix A) in Zerbolò (Italy, PV, 45°21′03.49″ N; 9°05′78.16″ E). It is a small strip of the Po Valley established as a Nature Reserve by the Italian State with the Ministerial Decree of 11 December 1973 of the Ministry of Agriculture and Forestry. The altitude is 65 m.a.s.l.. The flora is dominated by the trees *Quercus robur* (English oak), *Populus alba*, *P. nigra*, *P. canescens* (several species of poplar) and *Robinia pseudoacacia* (black locust), and tall shrubs *Corylus avellana* (hazel), *Crataegus monogyna* (hawthorn), and *Prunus padus* (pado). Low shrubs are sporadic, covering no more than 15% and rarely exceeding one meter in height. The most frequent shrubs species are *Ligustrum vulgare* (privet) and *Euonymus europaeus* (priest’s cap). The vegetation of the forest is referred to the alluvial oak-ulm types, which can be classified in the phytosociological association *Polygonato multiflori-quercetum robori* [31]. All the three sites are characterized by a sandy loam soil texture.

Soil samples were collected at a depth of 10 cm and placed in sterile polyethylene bags using a manual shovel cleaned with 75% ethanol. Each sample was obtained by mixing three sub-samples (20 g each) taken along a straight line 50 cm apart, randomly in each site. Specifically, four soil samples were taken from the Polluted AgroEcosystem (PAE), three from the AgroEcosystem (AE) and four from the Integral Natural Reserve “Bosco Siro Negri” (INR), for a total of 11 soil samples.

The soil samples were transported to the laboratory in refrigerated containers, sieved with a 2 mm mesh, removing roots and plant debris, and stored at a temperature of −20 °C. They were then used for physico-chemical, fungal counts and metabarcoding analyses.

### 2.2. Evaluation of Soil Plastic Fragments Counts

Pollution caused by macro and microplastics was evaluated in each study site.

The number of macroplastics (>5 mm) was assessed in 250 g of soil by counting the residues remaining in a 5 mm mesh sieve.

Microplastic count was performed using the oil extraction protocol developed by Crichton et al. [32]. A solution of 100 mL of filtered water and 50 g of soil was shaken by magnetic stirrer for 2 min. Then, 5 mL of canola oil was added to the soil solution and stirred for other 5 min to allow a better blend. The suspension was left to settle for 30 min, and the oil layer was collected by a funnel. The obtained oil was observed by stereomicroscope to count microplastic fragments (<5 mm).

### 2.3. Soil Physico-Chemical Analyses

Physico-chemical parameters of the sampled soils were determined by the Department of Earth and Environmental Sciences of University of Milano-Bicocca (Milan, Italy), according to the Italian standard protocols (DM 13/09/99). The measured parameters were: pH, organic matter, total nitrogen (TN), organic carbon (SOC), C/N ratio, plant-available phosphorous (P), calcium (Ca), magnesium (Mg), potassium (K), soil composition in sand, silt, and clay.

### 2.4. Evaluation of Soil Fungal Counts

To count the number of growing fungal colonies, soil samples were processed within 15 days from collection, using the Dilution Plate Technique [33], partially following the protocol of Landínez-Torres et al. [34]. Soil dilutions at 10^−3^ were prepared for four replicates of each sample, and 100 µL of these dilutions were spread on malt extract agar (MEA; Merck KGaA, Darmstadt, Germany) plates. Moreover, in order to estimate the number of fungal strains with strong abilities of degrading recalcitrant materials, fungal counts were also performed on humic acid agar (1 g of commercial humic acids, 3.26 g of Bushnell-Haas broth, 15 g of agar; Merck KGaA, Darmstadt, Germany) and lignocellulose agar (4 g of lignocellulose, 3.26 of Bushnell-Haas broth, 15 g of agar) plates [35]. Inoculated plates were incubated at 25 °C in the dark and observed every day for 2 weeks. The number of developed fungal colonies was expressed as CFU (Colony-Forming Units) per gram of soil dry weight.

### 2.5. Soil Microbial Communities Metabolic Profiling

Fungal functional diversity was studied using a Biolog EcoPlate^TM^ (Biolog, Hayward, CA, USA). It is a 96-well plate containing 31 different carbon substrates in three replicate sets and three negative controls, which can be divided into six categories: amines, amino acids, carbohydrates, carboxylic acids, phenols, and polymers (Appendix A). Biolog EcoPlates are useful to evaluate the functional diversity of soil microorganisms of soil based on the capacity of substrate utilization. Soil dilutions were prepared using 10 g of each sample with 100 mL sterile 0.85% NaCl solution and shaken horizontally for 30 min at 25 °C. Then, they were diluted in sterile saline solution to reach 10^−2^ dilutions, and 150 μL of the obtained samples was added into each well of Ecoplates. The plates were incubated at 25 °C for 96 h, and absorbance was measured at 590 nm after 0, 17, 24, 48, 65, 72, 90, and 96 h using a Bio Tek 800 TS microplate reader.

Biolog EcoPlates allowed us to calculate the average well colour development (AWCD) using the Equation (1) by Zabinski and Gannon [36]:(1)$$AWCD=\sum \left(C_{i}-R\right)/31$$
where *C_i_* is the absorbance of the single well corresponding to carbon source reaction, and *R* is the absorbance of the control well.

Moreover, the average well colour development of the six types of carbon sources (SAWDC) was calculated using the Equation (2) by Wang et al. [37]:(2)$$\sum (C_{i}-R)/n$$
where *n* is the total number of carbon sources in each category.

### 2.6. DNA Metabarcoding and Bioinformatic Data Analysis

DNA extraction and amplification was performed following the protocol by Nicola et al. [38]. The FastDNA^TM^ SPIN Kit for Soil (MP Biomedicals, Santa Ana, CA, USA) was used to extract DNA from 500 mg of each soil sample, according to the manufacturer’s instructions. The extracted DNA was quantified by a NanoDrop™ Lite spectrophotometer (Thermo Fisher Scientific Inc., Waltham, MA, USA), duly diluted, and stored at −20 °C until PCR amplification. The ribosomal ITS1 region (Internal Transcribed Spacer region 1) was chosen as a target for amplicon production, using the primers BITS and B58S357 [39] linked to Illumina adapters. The PCR reaction was performed following the protocol by Landínez-Torres et al. [34], with a cycling program which included an initial denaturation (95 °C for 3 min), followed by 25 cycles at 94 °C for 30 s, 58 °C for 30 s, 72 °C for 30 s, and final extension (72 °C for 5 min). The following steps of purification, amplification, and normalization of PCR amplicons were performed according to Nicola et al. [38]. The obtained product was loaded on the MiSeq System (Illumina, Inc, Illumina Way, San Diego, CA, USA, 92122.) and sequenced following the V3—300PE strategy. The bioinformatic analysis was performed by Qiime2 version 2020.2 [40], again following the protocol by Nicola et al. [38]. UNITE v.8.2 was used to associate the taxonomy to the obtained reads [41], following the classification by Tedersoo et al. [42]. Sequencing and bioinformatic data analysis were performed at BMR Genomics srl (Padua, Italy).

### 2.7. Statistical Analysis

Chemical data, CFUs and plastic counts and Ecoplates data were statistically analysed with the PAST software package, version 4.10 [43], using the ANOVA test with following Tukey’s pairwise test. Statistical analysis of the sequencing data was performed with the phyloseq R package, ver. 1.38.0 [44], following the protocol by Nicola et al. [38] on samples rarefaction at 90%, calculation of alpha diversity indices, beta-diversity multivariate analysis through Principal Coordinate Analysis (PCoA) on unweighted UNIFRAC distance matrix, PERMANOVA test and calculation of differentially abundant OTUs.

## 3. Results

### 3.1. Evaluation of Soil Plastic Fragments Counts

The counting of plastic fragments revealed a significantly higher number of total plastics in the PAE compared to AE samples and INR samples (ANOVA, *p* < 0.05). In particular, the number of microplastics (MP) present in the PAE was two orders of magnitude higher than that in the AE and three higher than that in the INR (Table 1). No plastic fragments larger than 5 mm were found in INR soil samples, while more than 5000 were detected in PAE soil.

### 3.2. Soil Physico-Chemical Analyses

Soil texture for the integral natural reserve (INR) and the Polluted AgroEcosystem (PAE) resulted as sandy loam, while the soil texture for AgroEcosystem (AE) was loamy sand, according to the USDA soil texture classification. The ANOVA statistical test detected significant differences in many parameters between INR and AE samples (pH, organic C, organic matter, total N, C/N and Mg). Indeed, INR samples had a significant higher concentration of organic matter, organic C, total N and a higher C/N ratio, which was coupled with a lower pH and Mg concentration. Moreover, PAE and AE samples differed significantly on Ca concentration (much higher in PAE samples) and K concentration (lower in PAE samples; Table 2).

### 3.3. Evaluation of Soil Fungal Counts

The evaluation of total fungal counts on MEA detected a significantly higher number of fungal CFUs in the integral natural reserve samples (INR) compared to the samples from the AgroEcosystem (AE) and the Polluted AgroEcosystem (PAE; ANOVA, *p* < 0.05). On the other hand, the counts on lignocellulose agar and humic acid agar detected a significantly higher number of CFUs able to grow on recalcitrant substances from the PAE samples compared to AE samples and INR samples (ANOVA, *p* < 0.05; Table 3).

### 3.4. Soil Microbial Communities Metabolic Profiling

The Biolog EcoPlate^TM^ was used to evaluate the metabolic profile of the three sites with different grade of plastic pollution. The average well colour development values (AWCD) were correlated with the incubation time (Figure 1). In the first 17 h, no change in well colour was detected in all samples, whereas a 24 h onset of activity was recorded in AE and INR sites but not in the PAE. After 65 h, AWCD increased slightly to a plateau and stabilized at 96 h in the AE and INR. Differently, the AWCD of the PAE grew much less throughout the incubation period, reaching a maximum value at 96 h significantly lower than the AWCD of the other two sites (*p* < 0.05).

SAWCD values were obtained after 96 h of incubation in order to evaluate metabolic preferences towards six different classes of carbon sources (Figure 2). The results showed similar performance for the INR and AE samples, favouring the use of phenolic compounds over the other categories tested. The trend in carbon sources utilisation was similar for all sites, although it was almost halved in PAE samples. In fact, the SAWCD values of the PAE were significantly lower than those of INR and AE values for each carbon sources category. Exceptions were polymers and phenolic compounds, whose differences were significant only for INR and AE, respectively (ANOVA, *p* < 0.05). Moreover, the SAWCD value associated with amines was almost zero in the PAE.

### 3.5. Soil Fungal Composition

The Illumina MiSeq sequencing of the DNA extracted from the soil samples resulted in 1,661,132 raw reads (approximately 147,242 ± 42,301 per sample). After filtering, denoising and merging steps, along with the elimination of chimeric sequences, 926,863 reads remained (approximately 81,680 ± 26,525 per sample). A total of 757 OTUs (Operational Taxonomic Units) was detected. The percentage of “unidentified” OTUs was relatively low, remaining around 0.4–4% depending on the sample.

The metabarcoding analysis detected the presence of nine different fungal phyla in the soil samples: *Mortierellomycota*, *Ascomycota*, *Basidiomycota*, *Mucoromycota*, *Monoblepharomycota*, *Rozellomycota*, *Aphelidiomycota*, *Kickxellomycota*, and *Basidiobolomycota* (Figure 3). The most abundant phylum was *Mortierellomycota*, with a relative abundance that ranged from 46% in AE to 52% in INR. Successively, there were *Ascomycota* (ranging from 23% in INR to 39% in AE), *Basidiomycota* (ranging from 11% in AE to 19% in INR) and *Mucoromycota* (ranging from 0.02% in AE to 5% in INR). The remaining phyla (*Aphelidiomycota*, *Kickxellomycota*, and *Basidiobolomycota*) were detected in very low abundances (<1% of the total reads). The percentage relative abundance of every taxon throughout this work, is calculated on the total of the reads.

#### 3.5.1. Phylum Ascomycota

Within the phylum *Ascomycota*, the metabarcoding analysis detected the presence of eight different classes (Figure 4A). The most abundant classes were *Sordariomycetes* (ranging from 6% in INR to 21% in AE), which is represented mainly by the orders *Hypocreales* and *Sordariales*; *Eurotiomycetes* (ranging from 9% in AE to 12% in the PAE), which is mainly composed by the order *Helotiales*; and *Dothideomycetes* (ranging from 0.5% in INR to 3% in PAE), which is mainly composed by the order *Pleosporales* (Figure 4B). The class *Sordariomycetes* was dominant in PAE and AE samples. Within this class, the family *Nectriaceae* was the most abundant one in PAE and AE samples (INR: 1.76%, PAE 8.56%, AE 8.23%), with a great amount of OTUs belonging to *Fusarium* (PAE: 2.68%; AE: 1.65%; INR: 0.00%). On the other hand, the ascomycetous composition in INR samples was mainly dominated by *Eurotiomycetes* and *Leotiomycetes*, with a particular abundance of the family *Aspergillaceae* (INR: 5.23%, PAE 1.58%, AE 1.55%). This abundance was mainly due to the presence of genus *Penicillium* (5.04% in INR).

In general, within the *Ascomycota*, 119 different genera were identified, the most abundant ones being *Fusarium* (4.6%), *Penicillium* (2.4%), *Furcasterigmium* (1.9%), *Talaromyces* (1.0%), and *Exophiala* (0.8%, Figure 4C).

#### 3.5.2. Phylum Basidiomycota

Within the phylum *Basidiomycota*, the analysis detected five different classes (Figure 5A). The most abundant ones were *Agaricomycetes* (ranging from 10% in AE to 17% in INR), represented mainly by the *Agaricales* and *Techisporales* orders; and *Tremellomycetes* (ranging from 0.4% in AE to 1.4% in INR), represented almost completely by the *Filobasidiales* order (Figure 5B).

Among the *Basidiomycota*, 51 genera were detected, and the most abundant were *Phaeoclavulina* (2.4%), present only in INR, *Parasola* (2.1%), *Russula* (0.9%), *Solicoccozyma* (0.8%), and *Alnicola* (0.7%, Figure 5C).

#### 3.5.3. Phylum Mortierellomycota

In this study, the *Mortierellomycota* phylum was the most abundant one, represented solely by the *Mortierellomycetes* class. Within this class, 15 different species were detected (Figure 6), all belonging mainly to the genera *Mortierella, Podila, Linnemania,* and *Gryganskiella*. The dominant species were *Podila humilis* (12%), which is present only in INR samples, *Mortierella alpina* (8%), *Podila minutissima* (5%), *Gryganskiella fimbricystis* (4%), and *Linnemania elongata* (3%). *Mortierella alpina, Gryganskiella fimbricystis,* and *Linnemania elongata* were not detected in INR samples.

#### 3.5.4. Phylum Mucoromycota

Within the *Mucoromycota* phylum, two different classes were detected: *Mucoromycetes* and *Umbelopsidomycetes*. In these classes, ten different species were recorded (Figure 7). The most abundant species was *Umbelopsis dimorpha* (1.65%), which is present only in INR samples, followed by *Rhizopus arrhizus* (0.05%), and *Gongronella koreana* (0.05%), which are both absent from INR samples.

### 3.6. Diversity of the Soil Mycobiota

The alpha diversity calculated for the fungal communities in the three different sites (INR, AE, PAE) was comparable, and no significant differences were detected in the indices of richness and evenness (Table 4, Pairwise Wilcoxon Rank-Sum test, *p* > 0.05).

The beta diversity, intended as the study of taxa composition within the three different sites, revealed differences among the fungal communities in the samples. Indeed, the PCoA (Principal Coordinates Analysis) based on an unweighted UNIFRAC distance matrix showed that the INR samples clustered together, distant from PAE and AE samples (Figure 8). This difference in the fungal community composition between INR samples, and PAE and AE samples was statistically confirmed by the permutational analysis of variance (PERMANOVA, *p* < 0.05).

The differential abundance of fungal species was calculated by differential expression analysis (DESeq2 R package) in order to assess if the fungal communities had species with significantly higher or lower abundances among the sites. Only the species of fungi with an abundance > 0.5% in at least one site were considered (Table 5). Considering the INR site, 11 fungal species were significantly more abundant compared to PAE and AE soil samples. On the other hand, 14 fungal species had a significantly higher abundance in PAE and/or in AE soil samples when compared to INR soil samples. Lastly, the comparison between PAE and AE fungal communities detected just two differentially abundant fungal species.

## 4. Discussion

The present work studied the composition of fungal communities and their metabolic activities in the soil of three different sites in northern Italy, where the concentration of macro- and micro-plastic was measured. Both abiotic and biotic analyses showed differences between the sites examined.

First of all, an evaluation of plastic fragments counts in the soil samples of each site was performed. The results showed that the samples from the Polluted AgroEcosystem (PAE) had a concentration of microplastics that was 100 times higher than the other two sites, reaching 13,200 ± 2466 microplastics/kg of soil. Similar microplastics (MP) concentrations were measured in urban and industrial soils [9] and in the derelict e-waste disassembling sites and polluted farmlands in China, where the concentration of microplastics exceeded 12,000 MP/kg [45,46]. On the other hand, the concentrations detected for the AgroEcosystem (AE; 160 ± 33 MP/kg) were lower than those reported for cultivated fields, especially those subjected to sludge applications in several areas of the world [47,48,49,50].

Alongside the quantification of the plastic presence, chemical and physical analyses of the soil were carried out in order to investigate the abiotic component in the three sites. It is commonly recognised that soil organic matter and organic carbon play a key role in soil quality, being a repository of nutrients and a crucial pool in the carbon cycle [51]. Together with pH, available P, and K, they are widely considered as soil quality indicators [52]. The high percentage of organic matter in INR was in accordance with what can be expected in an integral forest and indicates a high quality of the soil. On the contrary, AE resulted in impaired soil in accordance with the Wisconsin Soil Health Scorecard for agricultural and farm soil [53]. This difference in soil quality between the INR and AE was also confirmed by the statistically significant differences of other parameters such as of organic carbon, total N, and C/N. These values were particularly low in AE, similar to those found in fields after years of cultivation [54,55,56]. This could be due to its proximity to cultivated fields, which may have depleted the soil in the entire surrounding area.

The results of the pH assessment in the three sites showed a similar acidic nature of the soils, which was particularly pronounced in INR. Furthermore, it may be interesting to note that the values found in the PAE were similar to those obtained after the addition of microplastics to the soil in in vitro studies [57,58]. This alteration in pH of soil after microplastics augmentation could be attributed to the increase in soil aeration and porosity due to the presence of plastics fragments in soil [21,58] and to the alteration of the community of nitrogen-fixing bacteria [20], which would change the concentration of NH_4_^+^, increasing pH [59].

Analyses of the metabolic capacities of soil microorganisms in the three different sites were carried out. In particular, the utilisation of six different categories of carbon substrates was assessed using Biolog EcoPlate^TM^, evaluating the microbial physiological profiles and metabolic activities [60,61]. The results showed a general reduction in microbial communities’ metabolic activities in the PAE, reflecting the low oxidative capacity of microorganisms in polluted soil (AWCD). This reduction in the PAE was confirmed by SAWCD, even if the carbon sources preference was similar in the three sites. In particular, a preference for phenolic compounds was detected in all samples. Phenols represent one of the most abundant components in soils [62,63], and they are mainly degraded by fungi and bacteria thanks to the release of extracellular enzymes [64]. On the other hand, the poor amine utilisation could be due to the very high specificity of the metabolism to degrade these substances (putrescine in particular) and thus to use them as a carbon source [65]. Unexpectedly, even if the high concentration of plastics in the soil led to presume an adaptation of microorganisms in the degradation of polymeric substrates, the metabolic intensity of the polymer category was also lower in the PAE than in the other two sites. Although no other studies were found investigating the modification of the metabolic activity of microorganisms in plastic-polluted soils using Biolog EcoPlates^TM^, similar observations in the reduction in metabolic activity were reported in the presence of various kinds of pollutants [66,67,68,69]. Specifically, the AWCD of waste-activated sludge contaminated with heavy metal followed a similar trend as in the PAE, highlighting that sludge without heavy metals had stronger metabolic activities [68]. Likewise, a reduction in AWCD values was observed as soil radiation pollution levels increased, suggesting that radiation had great effects on microbial activities [69]. Therefore, these and other studies confirm that pollution can affect the metabolic efficiency and activities of microbial communities.

After the evaluation of the total microbial metabolic activity, the count of cultivable saprotrophic fungal community was performed. An interesting result showed that Colony-Forming Units (CFUs) growing on rich medium were statistically more abundant in INR soils, while CFUs growing on selective media (humic acids and lignocellulose) were more abundant in PAE soils. Even if humic acids and lignocellulose are commonly found in forest environments, a hypothesis could be that the cultivable saprotrophic fungal community of the PAE was more adapted to producing exoenzymes necessary for survival in a stressful environment [70,71,72]. Moreover, these data could be also affected by the different latency of the propagules; that is, highly competitive micromycetes in the PAE and AE may be overrepresented compared to slow-growing or latent species in INR.

The analysis of soil fungal community composition through metabarcoding delivered substantial results. The indices of richness and evenness were not significantly different in the three sites. Other in vivo studies of fungal and bacterial communities also reported a similar alpha-diversity pattern in relation to the presence of microplastics in soil [73,74]. However, a significant difference in beta diversity between the fungal taxa composition of INR soils and the other two areas was detected, revealing the presence of a distinct fungal community in the integral natural forest.

An interesting result was that despite the fact that the PAE had a much higher concentration of plastic in soil and a significantly reduced carbon metabolism with respect to AE, the adjoining non-polluted site, the fungal diversity indices and the fungal community composition changed only slightly. It is possible to hypothesize that fungi in natural soils can tolerate high concentrations of macro and microplastics, which do not prevent the presence of most species, but they hamper and slow down their metabolism, as shown by the Biolog Ecoplate^TM^ results. On the other hand, soil bacteria, especially in the rhizosphere, are greatly influenced by microplastics [20,75]. It was possible to see some differences between the PAE and AE especially at the genus level (Figure 4). Focusing on the most abundant ascomycetous genera, the PAE had a higher abundance of *Fusarium*, *Neocosmospora*, and *Magnaporthiopsis*, while *Arxiella* and *Furcasterigmium* were more abundant in AE. This is consistent with the respective niches, since *Arxiella* species are typically found in leaf litter [76], and *Furcasterigmium* species (with particular concern to *F. furcatum*) in soil, although they are likely to be endophytes and/or parasites in cultivated plants [77]. The genus *Fusarium,* together also with *Neocosmospora solani*, formerly known as *Fusarium solani*, is ubiquitous in soil and has a cosmopolitan distribution [78]. Most species live as saprotrophs, but others are important plant pathogens that are abundant in agricultural soils [79]. Some *Fusarium* strains were also reported as being able to grow on plastic surfaces [80]. *Magnaporthiopsis* is a soil-borne plant pathogen too, a root necrotroph attacking mainly grass, maize, and cotton [81]. This increased abundance in plant pathogens in the PAE along with a high concentration of plastic, as found also by Ren et al. [82], needs to be further investigated in different agroecosystems equally polluted. Regarding basidiomycetous genera, *Parasola* was especially abundant in the PAE, while *Mycenella* is abundant in AE. *Parasola* species are strictly saprotrophic in litter and soil; like many other *Coprinus*-like fungi, they are often found in very disturbed environments too, as well as on biological wastes from human activities, such as cattle litter [83,84]. Analogously, *Mycenella* species are widespread on plant residues and debris, although they generally require less disturbed conditions because they take much longer time than *Coprinus*-like fungi to colonize their substrate. Quaintly, *Mycena* species appear to be more represented in the PAE, although they are usually associated with ecological features which are very similar to *Mycenella* ones [85]. As for *Mortierellomycota*, an increased abundance of *Mortierella alpina* was detected. There is evidence that *M. alpina* has bioremediation potential. Indeed, it was able to degrade polycyclic aromatic hydrocarbons (PAH) in soil [86], and it was also among the fungi isolated from abandoned industrial deposits [87], indicating a strong resistance to polluted environments. Regarding *Mucoromycota*, the community in the AE was constituted solely by *Rhizopus arrhenius*, while in the PAE, *Absidia pseudocylindrospora*, *Absidia repens*, *Gongronella butleri* and *Umbelopsis vinacea* were also present. *G. butleri* is a fungus of interest because it showed great biocomposting potential, which was corroborated by the production of an abundant and diversified set of catabolic exoenzymes, including laccases, proteases and amilases [88,89].

Regarding the general taxonomical composition of the soil fungal communities in the analysed samples, the most abundant phylum was *Mortierellomycota* in all the three sites, where 15 different species were detected. These species, now belonging to different genera (*Podila, Linnemania, Gryganskiella, Dissophora and Entomortierella*), were all formerly part of the genus *Mortierella*. This genus is ubiquitous in soil, and it represents one of the ten most frequently recovered fungal genera in environmental sequencing projects [90]. Indeed, in the last decade, biomolecular techniques revealed that *Mortierella* is one of the most abundant filamentous fungal genus in the soils around the world, together with *Penicillium* and *Aspergillus* [38,91,92,93,94,95]. The high abundance of *Mortierellomycota* in the samples could be furthermore explained by the preference of these fungi for acidic soils, as already observed by Ning et al. [96]. Soil pH, in fact, ranged from 4.4 in INR samples (strongly acidic) to 5.3 in PAE samples and 6.0 in AE samples (acidic). Moreover, members of *Mortierellomycota* are commonly found in soils due to their ability to degrade cellulose and lignin [97,98,99,100,101]. Furthermore, many strains belonging to *Mortierella* are able to degrade toxic substances such as aromatic hydrocarbons, pesticides and dioxins [102,103,104]; thus, they often inhabit also polluted soils [105]. Thanks to this marked degradative potential and synthesis of melanin, members of *Mortierellomycota* have an important role in the C cycle, increasing soil C stock with the formation and retention of stable organic matter [104]. Moreover, several species of *Mortierella* can increase the bioavailability of P and Fe [106,107].

Ascomycota was the second most abundant phylum after *Mortierellomycota*, with the wide and diversified class of *Sordariomycetes*, and subsequently the *Nectriaceae* family, as dominant especially in PAE and AE soils. On the other hand, in INR samples, the family *Aspergillaceae* was particularly abundant due to the genus *Penicillium*. This genus is one of the most common saprotrophic fungi, ubiquitous in soil, with a preference for temperate climates [108]. Species of *Penicillium* are cellulolytic and frequently found in natural forest soil, where there is a great quantity of leaf litter [109], and they were already detected in a previous study of soil of the same area [110]. Regarding the other classes of *Ascomycota*, *Pezizomycetes* was more represented in the INR, which is consistent with the presence of both micro- and macroscopic species in *Pezizales*. Most of these species are soil saprotrophs even in disturbed conditions, including a minority that has evolved peculiar adaptation to extreme conditions, such as burnt soils [111]. However, evolved soils and less disturbed conditions are consistent with higher diversity in the *Pezizales* community as well as with the possible occurrence of species proposed to be mycorrhizals [112]. *Leotiomycetes* were mostly represented by the orders *Helotiales* and *Thelebolales*, whose respective taxonomic relationship is still “contentious” [113], as well as the possible accomodation of certain taxa in different orders, such as the *Leotiales* Korf and Lizon. Interestingly, no genus in *Leotiomycetes* is mentioned amongst the most abundant ones recovered in this study, although they provide a major compound in the overall diversity, as shown in Figure 2. This suggests that a remarkable variety inside *Ascomycota* occurs without any taxon featuring dominance dynamics, even when considering the soil community only (that cuts off most propagules from biotrophic species). Moreover, just like *Helotiales*, species in *Thelebolales* show a wide spectrum of trophic behaviours and habitat preference, ranging from coprophyly to opportunistic parasitism in animals to plant debris degradation in freshwater [114,115].

In the phylum *Basidiomycota*, the class *Agaricomycetes* is the most widely represented in all the three sites, which is followed by *Tremellomycetes*. This is consistent with the environmental matrix under examination (soil), since these two classes mainly include saprotrophs (or necrotrophs) and root symbionts [116,117,118]. Such a scenario is even clearer in the INR soils, consistently with the variety of trophic niches in the soil itself. Thus, the larger fraction of recovered *Agaricomycetes* belonged to *Agaricales*, with the genera *Alnicola*, *Arrhenia*, *Calvatia*, *Lepista*, *Leucoagaricus*, *Mycenella*, and *Parasola* as the most represented, and all are related to saprotrophism in soil and litter. *Arrhenia acerosa* is the only (presumably) mycorrhizic species in this set, coming from the PAE and AE only. A major fraction in *Agaricomycetes* was provided by *Trechisporales*; all the species in this order are involved in lignocellulose degradation and produce exiguous basidiomes on wood or coarse debris, but their diversity in soil is still largely cryptic [119]. The third most represented order was *Gomphales*, which was mainly due to the widespread *Phaeoclavulina*, which is another saprotrophic genus in litter and plant residues in woody areas (here, *P. decurrens* from the INR only). Moreover, besides macroscopic, pollution-sensible mycorrhyzal species, *Cantharellales* include several plant pathogens, which are here represented by *Ceratobasidiaceae* in the PAE and AE, besides a (saprotrophic) *Clavulina* in the INR. Considering the diversity of *Basidiomycota* found in this study, most members of this phylum are wood and litter saprotrophic fungi, especially in INR. This site has been periodically subjected to water flood and frequent rising of the groundwater level, since it is placed close to the Ticino River. This phenomenon contributes to the fact that INR results are poorly suitable for rich mycorrhizal communities, whereas the deadwood habitat is particularly developed [120,121,122].

Regarding the differentially abundant fungal species in the entire mycobiota among the three sites, a group of eleven species was found as significantly more abundant in INR soils. The predominant species of this fungal community was *Podila humilis*, which is a little studied fungus previously known as *Mortierella humilis* and isolated from forest soils in different areas of the world, such as China, Mexico, North Carolina, Norway, and Korea [123,124,125]. It is active in the degradation of plant remnants thanks to its cellulolytic and ligninolytic action, contributing to the forest C cycle [100,126]. The finding of *P. humilis* in an Integral Natural Reserve Forest in the Po Valley in Italy contributes to knowledge of the distribution of this species, and further studies should focus on its dominant role in the fungal community of this natural site. Three different basidiomycetes were more abundant in INR soils. Among these, *Russula* sp. and *R. ionochlora* are the only strictly mycorrhizal genus gaining a significant fraction. Indeed, *Russula* basidiomes are rare in cultivated areas/field sides and disturbed areas and often found scattered in the INR (personal communication). The other two species are the above-mentioned *Phaeoclavulina decurrens* and *Paralepista flaccida* both related to broadleaves litter.

On the other hand, *Mortierella alpina* and *Linnemania elongata* (formerly *Mortierella elongata*) were significantly more abundant in PAE and AE soils compared to INR soils. As already mentioned, *M. alpina* showed good bioremediation potential [86,87], while also *L. elongata* was able to degrade a plastic polymer manufactured by blending metallocene-catalyzed polyethylene (mPE) with unripe banana flour [127]. Moreover, *Fusarium concolor* and *Neocosmospora solani* (formerly known as *Fusarium solani*) were more abundant in PAE and AE soils. Both species are plant pathogens, *F. concolor* of cereal crops [128], *N. solani* of various vegetables, fruits and flowers, causing also damping off and root rot [129]. Analogously, the basidiomicetous species *Calvatia cyathiformis* was absent in the INR since it is related to meadows, grasses and fields, not to woody areas; as other epigeous gasteromycetes, it is assumed to be a saprotroph. Interestingly, it is considered a rare species in Italy, at least based on the detectability of visible basidiomes [130].

Only two fungal species were found as differentially abundant between PAE and AE soils, both with an abundance <1%. *Geastrum morganii* is another gasteromycete found by previous researchers to produce basidiomes in the INR; this saprotrophic species is related to sandy soils in woody or semi-woody areas, which is consistent with both the INR and the AE sites. *Dissophora globulifera* was present in PAE and INR soils, while it was completely absent from AE ones. *D. globulifera*, formerly known as *Mortierella globulifera*, is, as many other species belonging to the Mortiarellaceae family, a soil and litter fungus able to produce arachidonic acid and other fatty acids [131,132].

## 5. Conclusions

In conclusion, this work is a detailed description of soil fungal communities and their metabolic activities in three different sites in northern Italy, where macro- and microplastic concentration in soil was measured. Microbial carbon metabolism was the most responsive parameter, as it was significantly reduced in all the six different types of carbon sources in the Polluted AgroEcosystem, reflecting the low oxidative capacity of microorganisms in this soil, which is rich in microplastics. The analysis of the soil fungal community composition revealed a general presence of fungal taxa known for their degradation of plant residues in all three sites, with fungi more sensitive to disturbed soils present only in the Integral Natural Reserve. Moreover, these data contribute substantially to the knowledge on the distribution of *Mortierellomycota* in soil of northern Italy, whose ecological role in this habitat should be examined in depth. Future studies should continue the research on understanding the effects of microplastic pollution in soil, especially analysing possible changes in the microbial metabolome and nutrient cycling.

## Figures and Tables

**Figure 1 jof-08-01247-f001:**
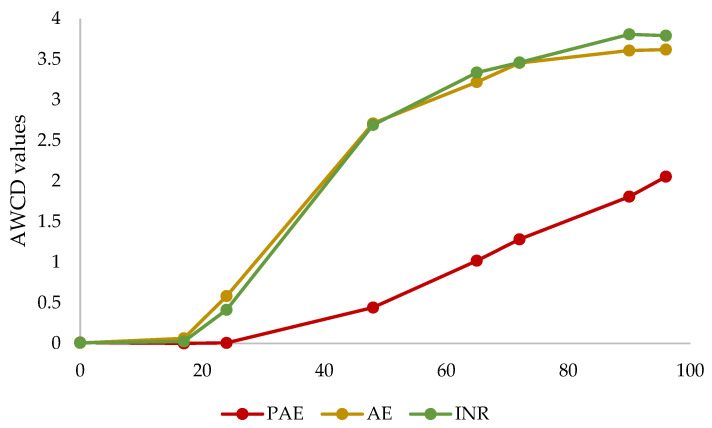
Variation in average well colour development (AWCD) values from 0 to 96 h of incubation in the Integral Natural Reserve (INR), AgroEcosystem (AE), and Polluted AgroEcosystem (PAE).

**Figure 2 jof-08-01247-f002:**
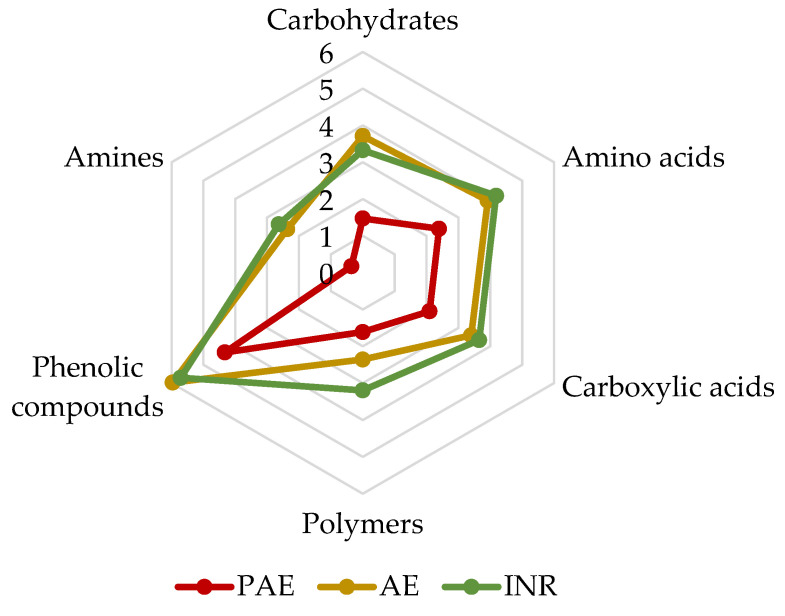
Average well colour development of the six types of carbon sources (SAWCD) after 96 h of incubation. In red Polluted AgroEcosystem (PAE), in yellow AgroEcosystem (AE), in green Integral Natural Reserve (INR).

**Figure 3 jof-08-01247-f003:**
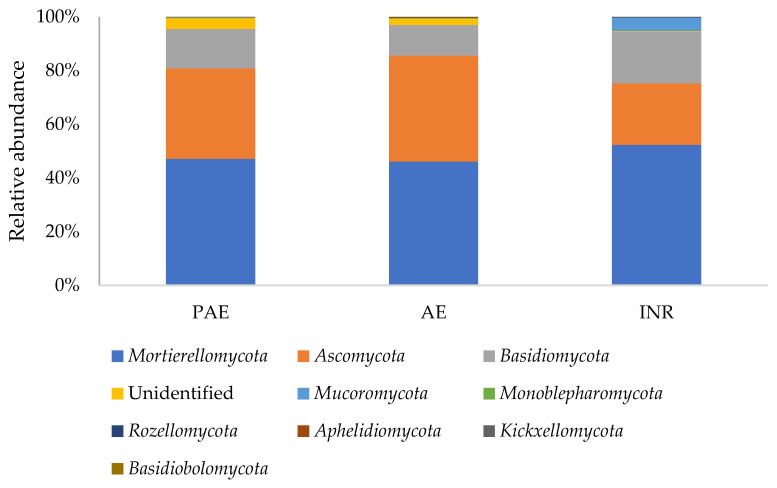
Average soil fungal relative abundances at phylum level of soil samples taken from the Polluted AgroEcosystem (PAE), the AgroEcosystem (AE), and the integral natural reserve (INR).

**Figure 4 jof-08-01247-f004:**
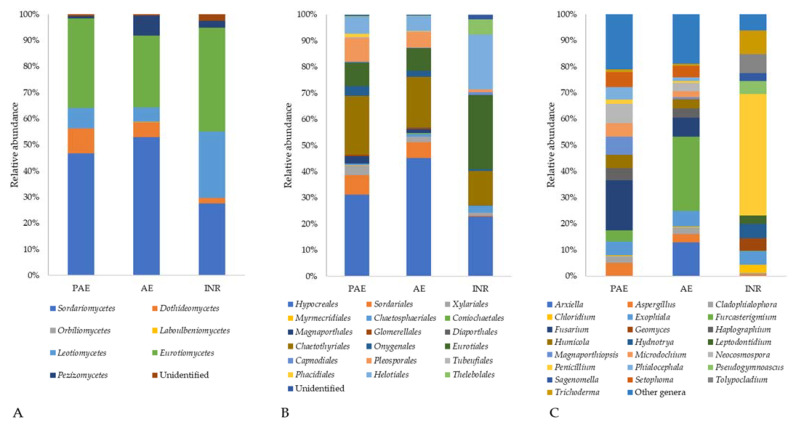
Relative abundances of *Ascomycota* classes (**A**), orders belonging to the most represented classes of *Ascomycota* (*Sordariomycetes*, *Eurotiomycetes*, *Leotiomycetes,* and *Dothideomycetes*) (**B**), and the most abundant genera (**C**) in soil samples taken from the Polluted AgroEcosystem (PAE), the AgroEcosystem (AE), and the integral natural reserve (INR).

**Figure 5 jof-08-01247-f005:**
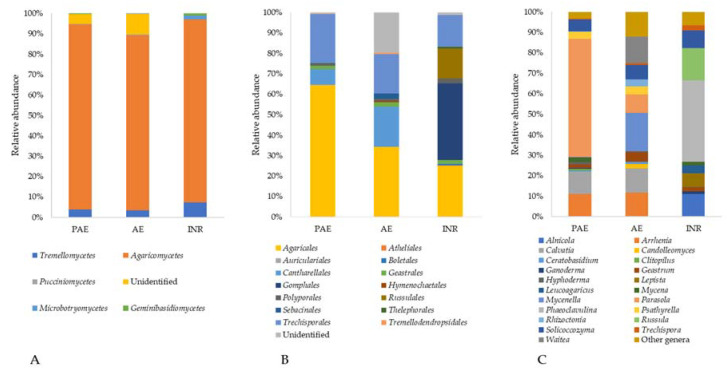
Relative abundances of *Basidiomycota* classes (**A**), of orders within *Agaricomycetes* (**B**), and most abundant genera (**C**) in soil samples taken from the Polluted AgroEcosystem (PAE), the AgroEcosystem (AE), and the integral natural reserve (INR).

**Figure 6 jof-08-01247-f006:**
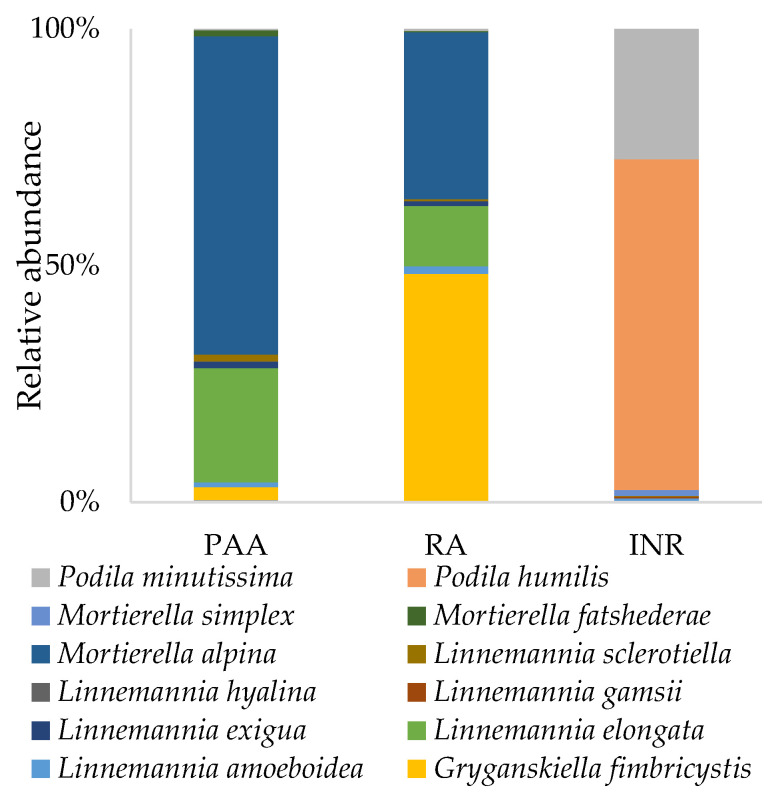
Abundance of the species belonging to *Mortierellomycota* in soil samples taken from the Polluted AgroEcosystem (PAE), the AgroEcosystem (AE), and the integral natural reserve (INR).

**Figure 7 jof-08-01247-f007:**
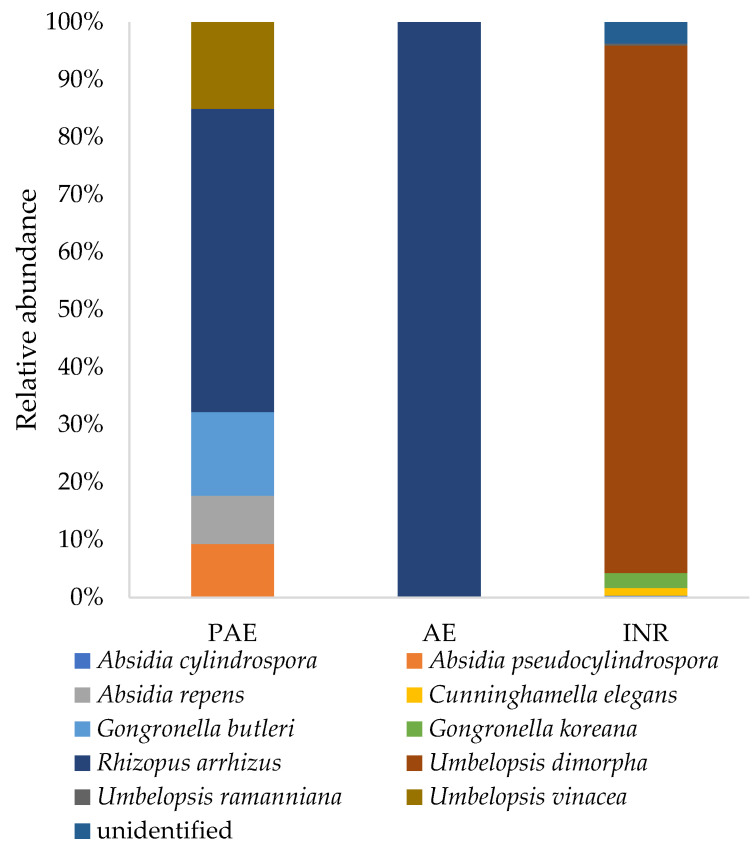
Abundance of the species belonging to *Mucoromycota* in soil samples taken from the Polluted AgroEcosystem (PAE), the AgroEcosystem (AE), and the integral natural reserve (INR).

**Figure 8 jof-08-01247-f008:**
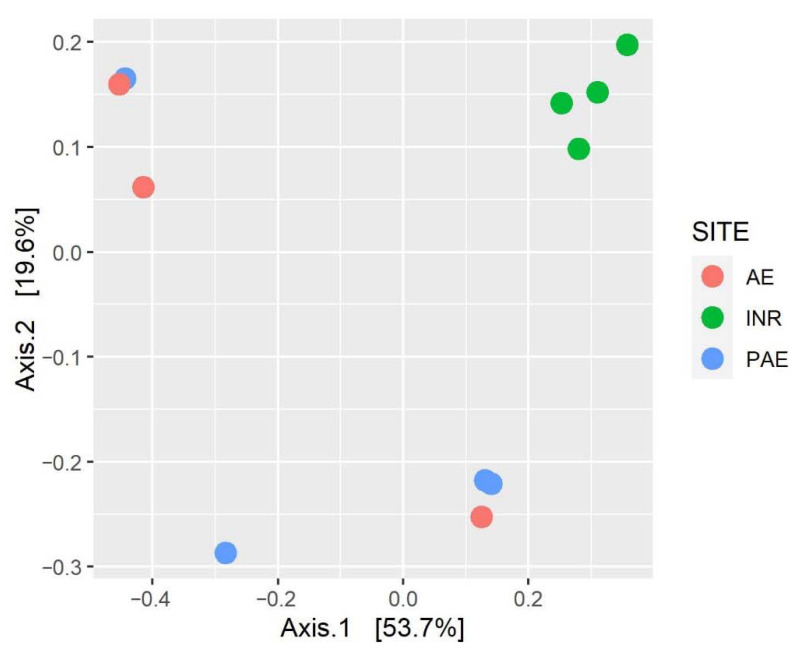
Principal coordinate analysis (PCoA) based on unweighted UNIFRAC distance matrix of Illumina MiSeq sequencing fungal data of soil samples taken from the AgroEcosystem (AE), the Polluted AgroEcosystem (PAE) and the Integral Natural Reserve (INR).

**Table 1 jof-08-01247-t001:** Plastic fragments counts in 1 kg of soil from the Polluted AgroEcosystem (PAE), the AgroEcosystem (AE), and the integral natural reserve (INR). The values displayed are the mean values ± standard deviation. Values with different letters in the same column are significantly different (*p* < 0.05, ANOVA test).

	Microplastics	Macroplastics	Total Plastic Amount
PAE	13,200 ± 2466 ^a^	5202 ± 230 ^a^	18,402 ± 2696 ^a^
AE	160 ± 33 ^b^	9 ± 11 ^b^	169 ± 44 ^b^
INR	60 ± 52 ^b^	0 ± 0 ^b^	60 ± 52 ^b^

**Table 2 jof-08-01247-t002:** Chemical analysis of soil samples taken from the Polluted AgroEcosystem (PAE) AgroEcosystem (AE), and the integral natural reserve (INR). The values displayed are the mean values ± standard deviation. Values with different letters in the same column are significantly different (*p* < 0.05, ANOVA test).

Site	pH	SOC	Organic Matter	TN	C/N	P	Ca	Mg	K
		%	%	g/kg		mg/kg	mg/kg	mg/kg	mg/kg
PAE	5.3 ± 0.0 ^ab^	3.1 ± 0.1 ^ab^	5.3 ± 0.2 ^ab^	2.97 ± 0.16 ^ab^	10.4 ± 0.2 ^ab^	24.0 ± 0.0 ^ab^	1423 ± 141 ^a^	137.6 ± 10.8 ^ab^	96.8 ± 10.2 ^a^
AE	6.0 ± 0.0 ^a^	1.4 ± 0.04 ^a^	2.5 ± 0.01 ^a^	1.43 ± 0.03 ^a^	10.1 ± 0.1 ^a^	30.0 ± 0.3 ^a^	857 ± 45 ^b^	143.7 ± 3.2 ^a^	272.5 ± 26.3 ^b^
INR	4.4 ± 0.0 ^b^	4.7 ± 0.1 ^b^	8.1 ± 0.2 ^b^	3.64 ± 0.08 ^b^	12.9 ± 0.4 ^b^	15.0 ± 0.5 ^b^	1041 ± 70 ^ab^	101.6 ± 10.0 ^b^	104.0 ± 16.1 ^ab^

**Table 3 jof-08-01247-t003:** Evaluation of total fungal counts (CFU) per gram of dry soil from the Polluted AgroEcosystem (PAE), AgroEcosystem (AE), and the integral natural reserve (INR) plated on MEA, humic acids, and lignocellulose. Values with different letters in the same column are significantly different (*p* < 0.05, ANOVA test).

	MEA	Humic Acids	Lignocellulose
PAE	5.0 × 10^5 a^	3.4 × 10^5 a^	3.7 × 10^5 a^
AE	5.2 × 10^5 a^	2.3 × 10^5 b^	1.5 × 10^5 b^
INR	5.8 × 10^5 b^	2.0 × 10^5 b^	1.8 × 10^5 b^

**Table 4 jof-08-01247-t004:** Richness and diversity indices based on Illumina MiSeq sequencing fata (mean ± standard deviation) of soil samples taken from the integral natural reserve (INR), the polluted AgroEcosystem (PAE) and the AgroEcosystem (AE).

	Observed Species	Shannon	Simpson
PAE	173.25 ± 38.78	3.98 ± 0.49	0.96 ± 0.02
AE	177.00 ± 73.82	3.91 ± 0.90	0.95 ± 0.04
INR	122.75 ± 9.11	3.29 ± 0.21	0.92 ± 0.03

**Table 5 jof-08-01247-t005:** List of fungal species with differential abundance in at least one site (differential expression analysis based on the negative binomial distribution, *p*-values < 0.05, adjusted by false discovery rate). Values with different letters in the same line are significantly different (*p* < 0.05, ANOVA test).

Fungal Species	Relative Abundance INR	Relative Abundance PAE	Relative Abundance AE
*Podila humilis*	33.51% ^a^	0.00% ^b^	0.00% ^b^
*Phaeoclavulina decurrens*	5.82% ^a^	0.00% ^b^	0.00% ^b^
*Penicillium bilaiae*	4.70% ^a^	0.00% ^b^	0.00% ^b^
*Russula* sp.	2.06% ^a^	0.00% ^b^	0.00% ^b^
*Lepista flaccida*	1.08% ^a^	0.00% ^b^	0.00% ^b^
*Tolypocladium album*	0.78% ^a^	0.00% ^b^	0.00% ^b^
*Hydnotrya tulasnei*	0.60% ^a^	0.00% ^b^	0.00% ^b^
*Mortierella simplex*	0.58% ^a^	0.00% ^b^	0.00% ^b^
*Phaeoclavulina decurrens*	0.55% ^a^	0.00% ^b^	0.00% ^b^
*Pseudogymnoascus roseus*	0.54% ^a^	0.00% ^b^	0.00% ^b^
*Geomyces auratus*	0.52% ^a^	0.00% ^b^	0.00% ^b^
*Mortierella alpina*	0% ^a^	15.22% ^b^	9.43% ^b^
*Linnemannia elongata*	0% ^a^	5.44% ^b^	3.42% ^b^
*Fusarium concolor*	0% ^a^	2.47% ^b^	1.44% ^b^
*Calvatia cyathiformis*	0% ^a^	1.06% ^b^	0.57% ^b^
*Neocosmospora solani*	0% ^a^	1.03% ^b^	0.71% ^b^
*Magnaporthiopsis incrustans*	0% ^a^	0.95% ^b^	0.15% ^b^
*Setophoma terrestris*	0% ^a^	0.79% ^b^	0.98% ^b^
*Tetracladium* sp.	0% ^a^	0.75% ^b^	0.2% ^ab^
*Microdochium novae-zelandiae*	0% ^a^	0.72% ^b^	0.52% ^b^
*Humicola olivacea*	0% ^a^	0.72% ^b^	0.78% ^b^
*Phialocephala bamuru*	0% ^a^	0.66% ^b^	0.32% ^b^
*Arrhenia* sp.	0% ^a^	0.64% ^b^	0.18% ^b^
*Haplographium debellae-marengoi* var. *equinum*	0% ^a^	0.63% ^b^	0.8% ^b^
*Furcasterigmium furcatum*	0.02% ^a^	0.59% ^ab^	6.31% ^b^
*Geastrum morganii*	0.15% ^a^	0% ^b^	0.19% ^a^
*Dissophora globulifera*	0.14% ^a^	0.04% ^a^	0.00% ^b^

## Data Availability

Data Availability Statement: The data presented in this study are openly available in the NCBI Sequence Read Archive (SRA, https://www.ncbi.nlm.nih.gov/sra, accessed on 30 September 2022) under the BioProject number PRJNA885759.

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
