# Peer review of "The Analysis of the Mycobiota in Plastic Polluted Soil Reveals a Reduction in Metabolic Ability"

_jof, 2022, doi:10.3390/jof8121247_

Round 1

Reviewer 1 Report

Refer to comments on paper

Author Response

Please, see the attachment. Note that the lines number refers to the new manuscript version with the visible corrections.

Reviewer 2 Report

Dear Authors, 

Great to read through your manuscript. The title was too general and read more like a review paper, covering inaccurate core results from the present study.

You presented their preliminary results on fungal diversity changes in three sites in Northern Italy, including an agricultural plastic-pulluted soil (PAE) and a non-plastic-pulluted soil (AE) and a natural reserve (INR). Generally, the three sites have some common features in fungal composition (a dominance of Mortierellmycota). However, it might be not suitable to define the sites as three different levels of plastic pollution soils, instead, the key interesting comparisons should made between PAE and AE, in terms of changes in fungal diversity, structure and composition in different levels.

Based on that, I would highly suggest the authors to refine their result between PAE and AE, and weaken the comparisons between the agricultural soils (PAE, AE) and the natural reserve.

Also, the figures and tables need to organize in a more integrated way, see below for detailed suggestions.

The information in Table 1 can be included as a supplementary file.

In table 3, please use the standard abbreviations: change C org to SOC, and N tot to TN; it would be great if authors could shorten the distance between column and lay the 8 soil parameters in a three-line table.

In Table 4, the multiple sign should be a  dot or *? please check the journals requirement.

Please remove the frame in Fig. 1.

In line 279, delete assemblage.

Add relative abundance in y-axis in Fig. 3. The same problem exists for other figures.

Please combine the Fig 4, Fig 5 and Fig. 6 as a new Figure 4, showing the Ascomycota information as three sub- bar graphs: Fig 4 A for the class, 4B for the highly abundent class, and 4C for the genera. The same suggestion for the figures 7, 8 and 9; as well as Figures 10 and 11.

Form the information in Table 2, Table 5 and Figure 12, we know that although the amounts of macro- and microplastics were all significantly higher in PAE than in AE, fungal diversity indexes and community structure were not obviously varied. So, what is your explanations on that? As far as I concerned, soil bacterial community might be more sensitive for microplastics, especially at plant rhizosphere.

Table 6, 7 and 8 can be combined as one Table, which would be more clearer for readers. Moreover, the fungal species name can be expressed more concise without naming person.

Author Response

Please, see the attachment. Note that the lines number refers to the new manuscript version with the visible corrections

Round 2

Reviewer 2 Report

Dear Authors,

Great job for considering all my comments in a careful way.

The manusript has imporved a lot with your efforts.

Best,